# The Contribution of the Minimal Promoter Element to the Activity of Synthetic Promoters Mediating CAR Expression in the Tumor Microenvironment

**DOI:** 10.3390/ijms23137431

**Published:** 2022-07-04

**Authors:** Yariv Greenshpan, Omri Sharabi, Ksenia M. Yegodayev, Ofra Novoplansky, Moshe Elkabets, Roi Gazit, Angel Porgador

**Affiliations:** 1The Shraga Segal Department of Microbiology, Faculty of Health Sciences, Immunology and Genetics, Ben-Gurion University of the Negev, Beer Sheva 84105, Israel; yarivg@post.bgu.ac.il (Y.G.); omrisha@post.bgu.ac.il (O.S.); kseniay@post.bgu.ac.il (K.M.Y.); ofranovo@gmail.com (O.N.); moshee@post.bgu.ac.il (M.E.); 2National Institute for Biotechnology in the Negev, Ben-Gurion University of the Negev, Beer Sheva 84105, Israel

**Keywords:** CARTIV, synthetic promoter, minimal promoter, chimeric antigen receptor

## Abstract

Harnessing immune effector cells to benefit cancer patients is becoming more and more prevalent in recent years. However, the increasing number of different therapeutic approaches, such as chimeric antigen receptors and armored chimeric antigen receptors, requires constant adjustments of the transgene expression levels. We have previously demonstrated it is possible to achieve spatial and temporal control of transgene expression as well as tailoring the inducing agents using the Chimeric Antigen Receptor Tumor Induced Vector (CARTIV) platform. Here we describe the next level of customization in our promoter platform. We have tested the functionality of three different minimal promoters, representing three different promoters’ strengths, leading to varying levels of CAR expression and primary T cell function. This strategy shows yet another level of CARTIV gene regulation that can be easily integrated into existing CAR T systems.

## 1. Introduction

Cancer cells originate from one’s own cells, hence the major difficulty in targeting them and not healthy tissue. In recent years there has been an ongoing hunt for new treatments that exclusively target cancerous cells. Immunotherapy’s main goal is to artificially navigate the immune system into attacking tumor cells precisely and specifically. One of the techniques in practice today involves chimeric antigen receptor T-Cells (CAR-T cells) [1]. Known as “Cars” in general, T or NK, aim to achieve cytotoxicity against specific antigen-expressing cells [2,3]. Several CAR T-cell therapies were granted FDA approval in recent years, all against CD19 in different B-cell malignancies [4,5]. Despite over 400 clinical trials currently in different stages, there is still no approved treatment for solid tumors, whether it be due to low efficacy or high toxicity mediated by the on-target off-tumor effect [6,7,8].

Many studies have shown innovative strategies to increase efficacy on one hand or reduce toxicity on the other. One of the methods to reduce toxicity is by the introduction of a suicide gene into the T cell. Here, an inducible caspase 9 (icas9) is introduced to the cells and is dimerized by the drug AP1903, thus forcing the cells to perform apoptosis [9,10,11]. Another approach is based on inhibition: a CAR against an antigen present primarily on healthy tissue is fused to an inhibitory receptor. That creates a “logic gate” forcing the CAR-bearing cell to choose to attack cancerous cells and not healthy tissue [12,13]. A different safety approach is limiting the CAR expression to the tumor site. The CARTIV method showed that it is possible to restrict CAR expression to the tumor by inducible promoters specific to the tumor microenvironment (TME) [14,15].

In order to increase treatment efficacy and to optimize CAR T cells treatment, there are several proposed modifications. These modifications, coined “armored cars”, use cytokines or ligands such as interleukin 12 (IL12) or 4-1BBL, inducible or constitutive, in order to try and increase persistence or cytotoxicity [16,17]. Here, a single promoter is required to drive the expression of both CAR and the auxiliary molecule, stressing the need to fine-tune the expression level to reach the desired therapeutic effect.

As we have shown, the CARTIV method is capable of restricting the transgene expression to the TME [15]. Our CARTIV synthetic promoters have two main parts: Promoter Response Elements (PREs) and a minimal promoter. The PREs can be customized to respond to specific TME signals, such as inflammatory signals or the tumor’s hypoxic environment. The second main component is the minimal promoter [14]. The minimal promoter, or core promoter, is a short sequence that allows for the formation of the initiation complex. The core promoter plays a critical role in the synthetic promoter properties, from its background expression, or leakiness, to its maximum potential induction.

Here we describe a way to adjust the promoter’s level of expression while maintaining its response to TME signals. Using different minimal promoters of varying transcription strength, we demonstrate the practical ability to adjust a given CARTIV promoter. We describe a series of synthetic CARTIV promoters having the same TME-responsive portion conjugated with different minimal core promoters, including the miniTK [18], miniCMV [19,20], and YB_TATA [21]. These core promoters show possible low background with weak maximal induction or high background with very high maximal induction. We further demonstrate how the expression levels are translated to surface expression of the CAR protein and change the overall CAR T-cell response to cognate antigen. Taken together, this study presents a simple direct method to modulate the overall expression levels of a given tissue-specific promoter and further suggests for threshold levels when translated into a functional surface receptor.

## 2. Results

### 2.1. Promoter Design and Constructs Used

A eukaryotic inducible synthetic promoter may consist of two main components, the promoter response elements (PREs), and the minimal promoter. The minimal promoter, or core promoter, is commonly a short sequence that allows the formation of a transcription initiation complex right at the transcription start site. We had previously developed CARTIV promoters [14,15] focusing on the PRE portion to gain response to common TME factors. We sought to further adjust expression levels by switching the minimal promoter portion. We narrowed down our work to three sequences representing a wide range of expression levels. The YB_TATA is a synthetic sequence developed by Benenson et al. [21] that proved to have a very low background expression [22]. The Herpes simplex thymidine kinase promoter (mini TK) is widely used [23,24]. It may suggest a compromise between background and maximum induction. The minimal cytomegalovirus promoter (mini CMV) is broadly used for strong levels of expression in many cell types [25,26,27].

We cloned either YB_TATA, miniTK, or miniCMV right after the G1K0.6H1 part of our recently published CARTIV [14,15], getting each minimal promoter with the very same PRE (Figure 1A,B). Our constructs drive either a reporter florescent RFP670 or a third generation Herceptin-based CAR [28]. To allow for the identification of transduced cells, all vectors harbored an independent constitutive ZsGreen fluorescent reporter, and all were packed in the phage lenti virus (LV) backbone.

### 2.2. G1K0.6H1 PRE-Combination Is Functional with All Three Minimal Promoters

In order to functionally test the promoters, we transduced HEK293T cells, which are responsive to hypoxia [29,30], and to a wide range of IFNγ and TNFα concentrations [15]. In order to correctly assess the promoter strength and background levels, we aimed to MOI of 0.3 or less—to gain single integration rather than multi-infection (data not shown). We first tested the series of promoters using the RF670 reporter construct. Cells were incubated with 125 U/mL IFNγ and 125 U/mL TNFα for 48 h and placed under hypoxic conditions for 18 h before FACS analysis. Thanks to the internal constitutive ZsGreen reporter, we could identify transduced cells (Appendix A). The promoter activity assessment considered two main parameters. First, the fraction of RFP670 positive cells (%RFP670^+^), gated of transduced cells and normalized to the total Zs^+^ population. Second, the geometric mean of the RFP670 fluorescent signal (MFI) of the Zs^+^ fraction.

As expected, the miniCMV promoter had the highest background expression, followed by the miniTK and a substantially lower background derived by the YB_TATA (Figure 2A). We also calculated the expression as a fold-increase over the unstimulated cells and noticed a very different pattern. Fold-induction found the miniTK to gain the highest relative expression (12.32 ± 0.87-fold change), followed by the YB_TATA (7.72 ± 0.17-fold change), and the miniTK, despite having the highest overall expression, showed the lowest relative increase (4.78 ± 0.44-fold change) (Figure 2B). We calculated synergism values (see Material and Method) to evaluate the coordinated added impact of the individual values. Both miniTK and the YB_TATA displayed a good synergism index of 1.69 and 1.37, respectively; however, the miniCMV had a synergism index of 0.89, probably due to high background expression (Figure 1A). Interestingly, both miniTK and miniCMV showed minute changes in the frequencies of RFP-positive (%RFP^+^) cells, ranging from 87.39% to 97.5% RFP670^+^ cells for the miniTK and 98.64% to 98.95% RFP670^+^ cells for the miniCMV (Appendix A). While some changes were noticed for the YB_TATA, ranging from 70.7% to 96.86% of %RFP670^+^ cells, for all three promoters most of the cells harboring the constructs (ZS^+^ cells) were RFP670 positive (Appendix A).

### 2.3. The Expression of a CAR on Cell’s Surface Is a Function of Promoter Strength

After confirming the functionality of all three promoters, we proceeded to CAR expression. The CAR used was a third generation Herceptin based CAR [28] with a MYC tag sequence between the 4D5 scfv and CD8 Hinge region (Appendix A) to allow for direct detection of the protein on the cell’s surface. HEK293T cells were transduced at 0.3 MOI and were allowed to recover for at least 72 h. The transgene expression was induced using 125 U/mL IFNγ and 125 U/mL TNFα for 48 h and placed under hypoxic conditions for 18 h. To assess for membranal CAR expression, cells were stained using an anti MYC antibody. The frequency of the reporter positive cells and the total MFI of the Zs^+^ fraction was measured.

In agreement with the reporter data, the miniCMV showed the strongest overall expression followed by the miniTK, whereas the YB_TATA had no detectable membranal CAR expression in all of the conditions tested (Figure 3A). Regarding fold change, we observed a different pattern with regard to the RFP670 reporter. The miniCMV had the highest fold change, 24.27 ± 1.57, whereas the mini TK had a 10.19 ± 2.86 fold (Figure 3B). The %Myc^+^ (i.e., CAR expressing cells) cells were also very different across the different minimal promoters. The background expression levels were low for the miniTK (1.36% ± 0.17% Myc^+^), while a substantial expression was recorded for the miniCMV (21.22% ± 3.21% Myc^+^). As expected, the miniCMV had the highest maximal response (92.45% ± 1.57% Myc^+^), followed by the miniTK (74.89% ± 2.86% Myc^+^) (Figure 3C). Both the miniTK and miniCMV showed synergistic behavior when calculated by gmean of the Zs^+^ fraction, 2.52 and 1.51 synergism index accordingly (Appendix A). However, CAR expression is inherently different from that of a reporter. Thus, it will be more appropriate to assess synergism by the percentage of CAR expressing cells rather than the gmean of the construct harboring fraction, the Zs^+^ cells. When calculating the synergism index from the %Myc^+^, The miniTK was the only promoter to show a synergistic behavior with a synergism index of 3.56 (Figure 3C). Taken together, these data support the possibility that the membranal expression of a CAR has a threshold behavior with regard to the promoter’s strength.

### 2.4. All Three Promoters Are Responsive to Combination Stimuli in Human Primary T Cells

Next, we proceeded to human primary T-cells which are clinically relevant to CAR-based treatments. Primary cells were grown in RPMI media 10% human serum and transduced with retronectin and spinfection as described in Methods. Following transduction, cells were rested for two days. Next, T cells were stimulated with 500 U/mL of IFNγ and TNFα for 48 h, and treated with hypoxia for the last 18 h. The human primary T cells manifested a substantial response to the combined stimuli of hypoxia and TNFα as well as to the single-factor stimulation in all three promoters for the reporter construct. Notably, there was no substantial IFNγ response, due to the autocrine secretion by these cells ex vivo [15]. As with the HEK293T cells, the miniCMV had the strongest total response followed by miniTK, and a substantially weaker response by the YB_TATA (Figure 4A). The fold change of the miniTK was the highest of the three promoters (3.99 ± 0.34-fold change), whereas the miniCMV and the YB_TATA showed similar fold change levels (2.30 ± 0.37 and 2.49 ± 0.19 accordingly, not significant) despite the miniCMV displaying 8.69 times stronger maximal response (Figure 4B). As for the %RFP670^+^ cells, all three promoters had a stimuli dependent response with varying levels of maximal and background expression. The miniCMV had high background expression levels (79.6%RFP670^+^) and a high maximum response (95.04%RFP670^+^). The YB_TATA had the lowest background level (6.38%RFP670^+^) and lowest maximum response (24.03%RFP670^+^). The miniTK background levels were in between those of the miniCMV and YB_TATA (26.09%RFP670^+^), as well as the maximum response (75.17%RFP670^+^) (Appendix A). Taken together, the data indicate that all three promoters are functional and responsive to the various inducing signals of the CARTIV G1K0.6H1 elements in human primary T-cells.

### 2.5. CAR-T Response Could Be Fine-Tuned by the Promoter Driving the CAR Expression

To test for possible benefits in employing different promoters, we proceeded to testing the membrane CAR expression and CAR mediated T cell function. TNFα stimulation had a week effect on CAR expression, while a substantial increase in expression was recorded following hypoxia and a synergistic response following the double stimulation for the miniTK (Appendix A). The miniCMV had high background expression levels compared to the miniTK and a slightly higher maximum expression when compared to the miniTK, (Figure 5A). The YB_TATA had no detectable membrane CAR expression in the tested conditions.

To assess for the maximal potential activation levels, cells were incubated on OKT3 coated wells (for robust anti-CD3 activation). Cell activation was measured using a CD107a degranulation assay. The percentage of degranulating cells was calculated as the fraction of CD107a^+^ cells from the ZsGreen^+^ population (Appendix A). Between all three promoters no significant difference was recorded for the background degranulation levels, nor for the maximal response levels in the ZsGreen^+^ population (Figure 5B).

The YB_TATA had no detectable degranulation in the ERBB2 coated well for stimulation conditions, probably due to lack of membranal CAR expression. For the miniTK there was no increase in degranulation levels following IFNγ, TNFα or the combination of the two. Only after adding the hypoxic conditions was an increase in degranulation recorded (Figure 5C). Despite the fact that TNFα as a single stimulus led to a slightly higher CAR expression, it did not lead to an increase in degranulation. A combination of TNFα and hypoxia showed higher degranulation levels then the hypoxic conditions alone, suggesting the synergistic expression by the promoter. The mini CMV promoter had higher basal degranulation levels than the mini TK (9.83% ± 1.36% vs. 4.10% ± 2.18%), probably due to basal CAR expression by the promoter. Here a slight increase in degranulation was recorded in response to TNFα as a single stimulus (9.83% ± 1.36% for non-stimulated versus 16% ± 1.98% for the TNFα alone) as well as for the hypoxia- TNFα combination (Figure 5C). It should be noted that all three T cell lines have similar maximum and basal CD107a expression (Figure 5B). Taken together, the membranal CAR expression and the degranulation data support the possibility to fine tune the CAR expression and by that limiting or enhancing the CAR-T response as needed.

## 3. Discussion

In this study we compared three minimal promoters using the CARTIV platform to characterize an optimal minimal promoter for inducible CAR expression. Given the importance of CAR density on the effector cell membrane [31], the ability to fine tune CAR expression will provide essential improvement. The modularity of switching parts of the synthetic promoter truly presents a useful toolbox for multiple usage. Gaining a sweet-spot intensity of CAR expression will allow for the balancing of tumor-killing and toxicity providing effective therapy to more and more patients.

The minimal promoters used in this study were all previously reported [18,20,21]. Being short sequences (60–25 bp), these are very easy to work with: simply order an oligo with the sequence of interest and flanking restriction-sites, or any other mode of cloning, such as gateway [32] or Gibson Assembly^®^ [33]. Moreover, the ability to work with a relatively short oligo opens an intriguing option to further refine such minimal promoters, just as we recently did with our CARTIV promoters [14]. Considering the known core transcription-initiating complex, and the structure of the protein-DNA complex, one may pick a few key nucleotides and change them to gain variations over the basic minimal promoter. Our data, showing easy mix-and-match functional construction of minimal promoters with PRE-portions such as the CARTIV used here, provide great opportunities for further improvements of synthetic promoters.

Our results suggest that the CARTIV promoter response element remains functional when coupled with all three minimal promoters. Pronounced RFP670 reporter expression was observed with the minimal CMV having the highest background of the three and the YB_TATA having very low background, only slightly higher than background levels. Normalization of each mini-promoter reveals that, despite overall highest expression levels by miniCMV, it is the miniTK that gains the best relative-induction and synergism. We must stress that gaining various patterns bring new options, and this may prove useful. Having substantial background levels may favor basal-levels of receptor signaling, supporting cellular viability and proliferation [34]. In other situations, minimal or no background might be needed to reduce toxicity altogether [35]. Intriguingly, the expression of an actual CAR, in contrast with the fluorescent reporter, showed critical differences already on the HEK293 cells. The YB_TATA failed to reveal detectable surface-proteins, miniTK was significantly reduced, and the miniCMV gained the best normalized-expression. Nevertheless, miniTK remained as best synergistic-partner. The difference between the internal fluorescent reporter and surface CAR protein are most likely the result of additional regulation of protein expression and processing [36,37]. This suggests yet another option for posttranslational regulation of CAR proteins and its different components, which is of significant interest and has been intensively studied recently [38,39,40].

Primary human T cells are currently the main type of immune effectors for clinical utilization [41,42]. Interestingly, several studies have suggested possible advantages of gamma-delta T cells [43,44], yet we keep with the more common and currently approved alpha-beta T cells [45]. Clearly, our CARTIV platform, with various PRE and mini-promoters, can easily fit with virtually any immune effector. We first examined the constructs with the fluorescent reporter due to the higher sensitivity. Data by and large resemble the earlier data with HEK293, confirming the potent activity of these promoters in primary human cells in vitro. Notably, the functional expression of CAR protein on the surface of primary T cells again resembles the earlier data with the cell line, confirming the validity of earlier screens and demonstrating very limited or even no expression of the CAR without stimulation. The functional degranulation, measured by CD107 expression, suggests yet another level of regulation. YB_TATA gained no increased degranulation, with agreement to the undetectable surface CAR. The normalized maximal increase by miniTK was inferior to miniCMV, in contrast to the earlier reporter expression. This is most likely due to some threshold-levels required for activation of the T cells [46,47]. Hence, despite the earlier notion of the miniCMV having relatively high background and less of a relative, normalized activity, when measuring the later effect over cellular activity it appears to benefit the overall expression. It is possible to achieve stronger expression levels and reduce the background by using an “indirect” expression circuit. For instance, the CARTIV miniTK promoter could be used to drive the expression of the reverse tetracycline-controlled trans-activator (rtTA), while the desired transgene will be expressed from the tetracycline response element (TRE). This could lead to external control over the expression by withdrawing the doxycycline while having high expression levels in the “on stat” from the strong TRE promoter.

Taken together, our data present the robust modular ability to change synthetic promoters. The option that response to TME can be separately designed and tuned, while the very short mini-promoter at the transcriptional start site can be easily flipped. Gaining the adequate level of expression of CAR, or other effector immune effector genes, will provide for better engineering of CAR-T and other immune cells to treat various types of tumors in more patients.

## 4. Materials and Methods

### 4.1. Cloning

The G1K06H1 Promoter response element was synthesized upstream to a mini TK minimal promoter as described elsewhere [15]. The minimal CMV promoter was synthesized as two ss-DNA oligos (hylabs, Rehovot) and annealed and cloned downstream to the G1K06H1 response element. The YB_TATA minimal promoter was added by whole plasmid PCR (Takara, CA, USA) phosphorylation and ligation (NEB, MA, USA).

### 4.2. Tissue Culture

HEK293T were grown in DMEM (Gibco, MA, USA) containing 10% serum, pen-strep, HEPES, L-glutamine, non-essential amino acids and sodium pyruvate (all from Biological Industries, Beit Haemek, Israel).

### 4.3. Human Primary T Cells Culture and Infection

Human primary T cells were extracted, cultured and infected as described extensively elsewhere [15]. Briefly, 7 mL of blood was taken from a healthy consenting donor and mononuclear cells were separated on a Ficoll gradient. The interphase was collected, washed and plated in RPMI 10% human serum, 100 U/mL rhIL2 and 50 ng/mL anti-human CD3 OKT3 (Biolegend, CA, USA). After 48 h, cells were collected, washed and re-plated in RPMI 10% human serum, 300 U/mL rhIL2 and no anti-human CD3 OKT3. Cells were infected using RetroNectin (Takara, CA, USA) coated plates followed by LV spin loading according to the manufacturer’s recommendations.

### 4.4. Lenti Virus Production and Concentration

LVs were produced in HEK293T cells cultured in DMEM 10% FBS (Biological Industries). Cells were grown to 90% confluence in 10 cm plates and were transfected with 10 µg of pHAGE2 vector and 3 µg of the packaging plasmids tat, rev, hgpm2 and vsvg in a mass 1:1:1:2 ratio. The transfection was performed using jetprime^®^ transfection reagent (Polyplus, France) according to the manufacturer’s recommendations. Media was added for the following two days and on the third day LV containing media was collected and 0.45 µ filtered and stored in −80 °C or concentrated by ultracentrifugation for 90 min at 17,000 RPM 4 °C. LV pellets were suspended in DMEM 10% FBS, aliquoted and stored in −80 °C.

### 4.5. Hypoxic Conditions

For hypoxic condition simulation, a hypoxic chamber was used. The experiment plate was placed in the chamber and the chamber was sealed. A gas mixture containing 5% CO_2_ 0.3% O_2_ and 94.7% N_2_ at 20 L/minute for three to five min was applied to the chamber. The chamber was sealed and placed at 37 °C for 16–20 h.

### 4.6. CARTIV Promoter Activity Assay

HEK293T cells plated at 1 × 10^5^ per well in a 96 well Flat bottom plate; human primary T cells at 2 × 10^5^ in a 96 well U-shape plate. Cytokines IFNγ and TNFα added to final concentrations of 500 U/mL. Hypoxia was induced for the last 16–20 h of the experiment. Cells were harvested, washed once in PBS 2% FCS, suspended with DAPI 1 µg/mL and FACS measured using a Beckman Coulter^®^ Cytoflex™ flow cytometer. Data were analyzed using Kaluza™ or cytexpert™ software. The normalized reporter expression was calculated by: $\frac{single\;stimulation\;MFI}{unstimulatd\;MFI}$. The synergism index was calculated by: $\frac{\left[IFN\gamma \;MFI\right]\;+\;\left[TNF\alpha \;MFI\right]\;+\;\left[Hypo\;MFI\right]}{\left[IFN\gamma \;MFI\;+\;TNF\alpha \;MFI\;+\;Hypo\;MFI\right]}$.

### 4.7. CAR Expression Assay

Cells harboring the promoters were induced as described above and harvested. Cells were washed once in PBS containing 2% FBS and stained using anti Myc tag antibody (Millipore) and incubated on ice for 1 h. Cells were washed once and stained using a secondary anti mouse APC (Jackson) and incubated on ice for 30 min. Cells were washed twice and suspended with DAPI 1 µg/mL and FACS were measured using a Beckman Coulter^®^ Cytoflex™ flow cytometer.

### 4.8. T Cell Functional Assay

Primary human T cells were plated at 5 × 10^4^ on 1 µg/mL ERBB2 (R&D) coated 96U plates together with anti-human CD107a APC (Biogems, Rehovot, Israel) and incubated for 4 h in a tissue culture incubator. After 4 h, cells were washed once and stained using anti human CD107a APC (Biogems, Rehovot, Israel) for 30 min on ice. Degranulation was assessed by FACS as above.

### 4.9. Statistics and Reproducibility

A paired one tailed *t*-test was performed where applicable.

## Figures and Tables

**Figure 1 ijms-23-07431-f001:**
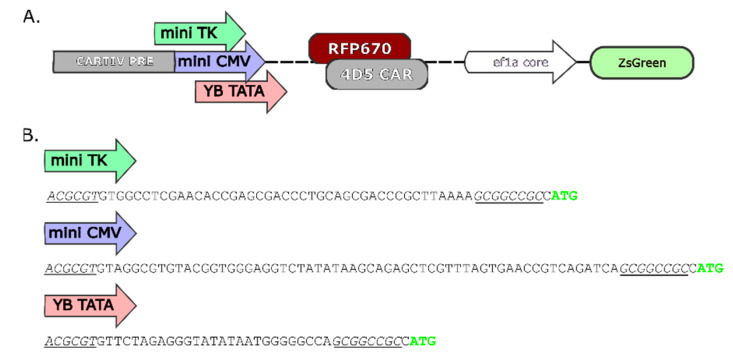
The different minimal promoters cupelled to the CARTIV promoter response element and the vectors used: (**A**) Schematics of the CARTIV synthetic promoters tested and the vector used. (**B**) Sequences of the three minimal promoters tested. In italic and underlined are the mluI and notI restriction sites, respectively, in light green the ATG start codon.

**Figure 2 ijms-23-07431-f002:**
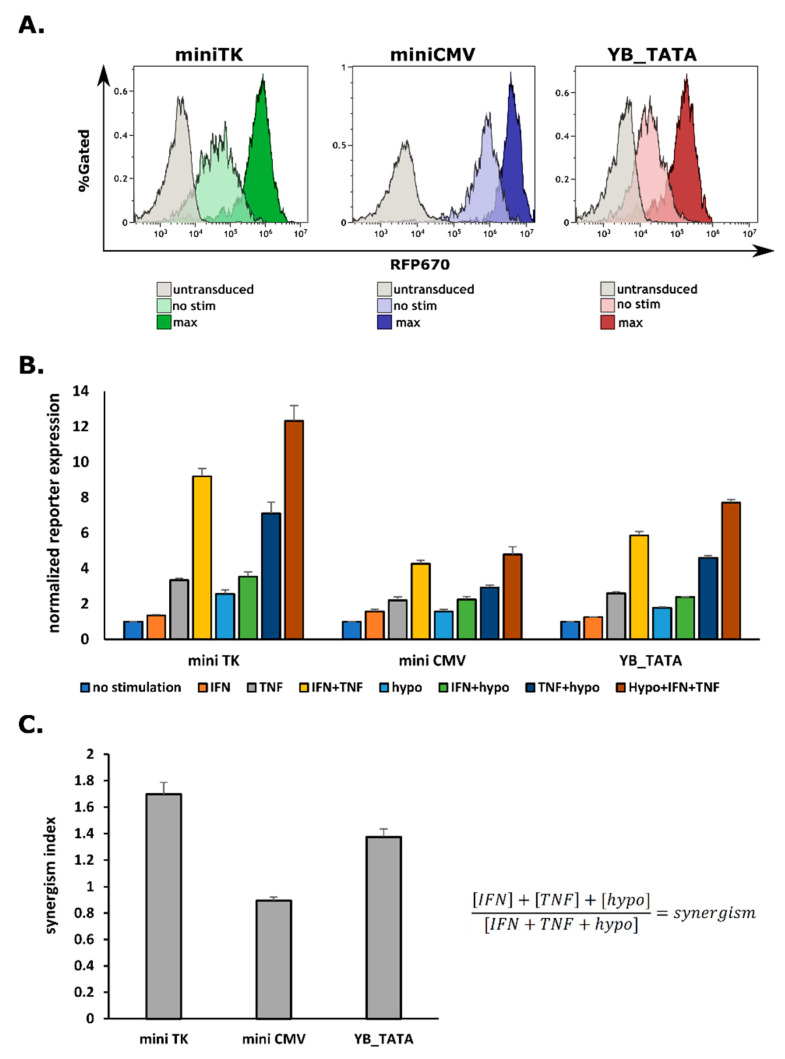
All three minimal promoters are functional when combined with the G1K0.6H1 CARTIV response element: HEK293T cells were infected using lentiviral vectors with RFP670 under the control of the indicated minimal promoter combined with the G1K0.6H1 CARTIV PRE and a ZsGreen reporter controlled by the ef1α core promoter. Data shown are ZsGreen-positive, single-discriminated, and DAPI-negative results. (**A**) Representative FACS plots of the indicated promoter’s background expression and maximum induction. (**B**) Reporter expression fold change geometric mean (gmean) average of triplicates from the ZsGreen positive fraction; error bars indicate standard deviation. (**C**) Synergism index of for the tested promoters and the formula used to calculate it.

**Figure 3 ijms-23-07431-f003:**
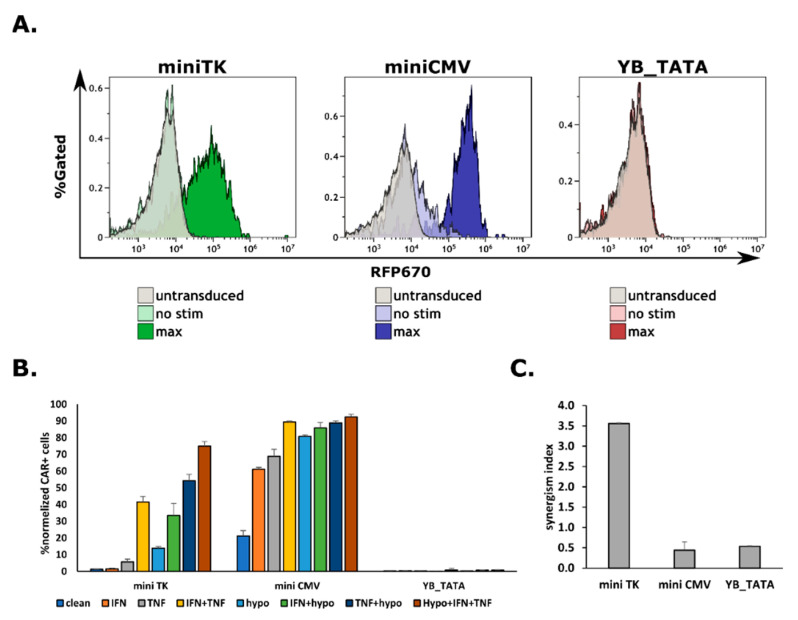
CAR expression under the different CARTIV promoters exhibits a threshold behavior: HEK293T cells were infected using lentiviral vectors with Herceptin based CAR under the control of the indicated minimal promoter combined with the G1K0.6H1 CARTIV PRE and a ZsGreen reporter controlled by the ef1α core promoter. (**A**) Representative FACS plots of the indicated promoter’s background and maximum induction CAR expression. (**B**) The %CAR+ cells normalized to the ZsGreen positive cells. Average of triplicates, error bars indicate standard deviation (**C**). The synergism index of for the tested promoters when driving CAR expression. (**C**) The synergism index of for the tested promoters in the CAR constructs.

**Figure 4 ijms-23-07431-f004:**
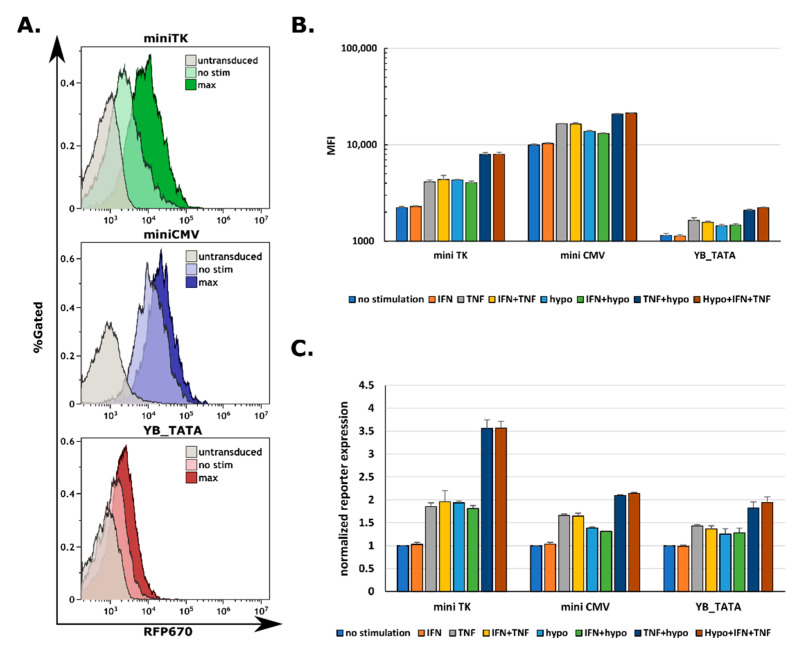
An additive response is observed in human primary T-cells for all three minimal promoters: (**A**) Representative FACS plots of the indicated promoter’s background and maximum induction rate of the RFP670. (**B**) Average from triplicates of MFI of the ZsGreen positive fraction; error bars indicate standard deviation. (**C**) Reporter expression fold change, average of triplicates from the ZsGreen positive fraction, error bar indicates standard deviation c. The %RFP670^+^ cells normalized to the ZsGreen positive cells. Average of triplicates; error bars indicate standard deviation. Data shown are ZsGreen-positive, single-discriminated, and DAPI-negative.

**Figure 5 ijms-23-07431-f005:**
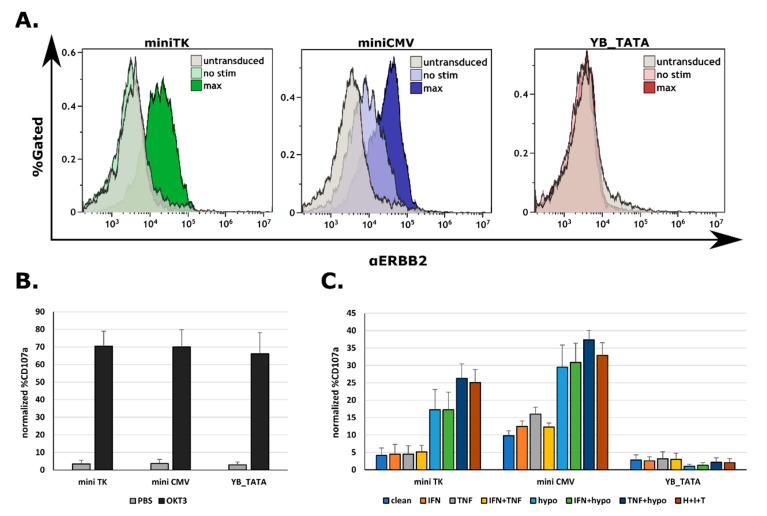
Degranulation against CAR cognate antigen shows a threshold behavior as a function of promoter strength: At 72 h following infection, the human primary T-cells were treated for 48 h with the indicated cytokines (250 U/mL for each cytokine) and for 18 h under hypoxic conditions. Cells were then incubated for 4 h with PBS, anti CD3 OKT3 or ERBB2-FC recombinant protein coated wells. Cells were than washed and stained using CD107a APC or a recombinant ERBB2-Fc. (**A**) Representative FACS plots showing membranal CAR expression by the indicated promoter in the non-stimulated and fully stimulated condition. (**B**) Maximum and background degranulation of human primary T-cells transduced with the indicated promoter controlling the expression of the Herceptin based CAR, showing only the Zs^+^ fraction. (**C**) Average from triplicates of the CD107a^+^ cells from the ZsGreen positive fraction, normalized to the total %ZsGreen positive cells in the indicated treatment and specific promoter.

## Data Availability

Data are contained within the article or Appendix A.

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
