# Peer review of "The Contribution of the Minimal Promoter Element to the Activity of Synthetic Promoters Mediating CAR Expression in the Tumor Microenvironment"

_ijms, 2022, doi:10.3390/ijms23137431_

Round 1

Reviewer 1 Report

CAR T based Immunotherapy has grown tremendously in importance, but there are still obstacles to be overcome, roughly these are on-target off-tumour effects and exhaustion of effector cells. Inducible receptor expression can contribute to improve therapeutic efficiency. One concept is a promoter design which is reactive to the immune suppressive tumour microenvironment called CARTIV, recently presented (Greensphan et al. 2021, Sharabi et al. 2022).

In their manuscript the authors show the modification of CRTIV by testing core promoter elements for their ability to modulate the expression ratio (stimulated vs. unstimulated). While the therapeutic relevance of a switch function with low background and efficient expression after stimuli is evident, the modulation ability by fixed structure may currently lack an applicable concept, but from technical point of view the report shows new contents and is interesting therefore.

In addition to the recently reported optimisation of the PRE (serial order of the sensitive elements and sequence optimisation), miniTK, miniCMC, and YP_TATA core promoters were tested and showed differences with respect to CAR background expression and CAR expression after stimulation, and relative expression increase.

Overall the manuscript is of good quality and well organized. However, the text should be carefully checked for missing citations (see p5), missing lines (see p5), formatting (see P4), and spelling (see p7).

Author Response

We thank the reviewer for his kind remarks and keen observation. All missing citations were inserted correctly (see pages 5,6 and 10 in track changes). The missing line on page 5 has also been aligned correctly (see page 5 in track changes). All formatting and spelling issues have been addressed throughout the manuscript.

Reviewer 2 Report

The manuscript describes the establishment of CAR-driven promoters in cells. It is interesting and well written. The experiments are well described. A few stylistic issues in the figures should be corrected. Thus, I recommend acceptance after minor revision:

Figure 2: Please specify the abbreviations in the caption. Please correct ´´… fold change  , gmean …´´ and ´´promotors´´. The same for Figures 3, 4 and 5.

Author Response

We thank the reviewer for his keen and precise observations. The abbreviations have been specified in all the indicated figure legends, as well as all spelling and formatting issues were addressed throughout the manuscript.